# Multi-Task Perception in Unstructured Environments: Anti-Degradation Complementary Learning and SAMEnhancer

## Abstract

While autonomous driving perception has advanced significantly in structured environments, unstructured environments still present major challenges due to the complexity of traffic participants and irregular road conditions. This paper focuses on addressing these challenges through multi-task perception, targeting drivable area segmentation and object detection in unstructured environments. A key issue in existing datasets for unstructured settings is the non-overlapping annotation of images across different tasks, which limits the efficiency of data utilization. To tackle this, we propose Anti-Degradation Complementary Learning (ADC learning), a semi-supervised approach that allows different tasks to share knowledge across unlabeled data, thereby maximizing the use of available image information. Additionally, we introduce SAMEnhancer, which integrates the Segment Anything Model (SAM) to improve segmentation quality by combining the semantic specificity of network training with the coherence of SAM's segmentation. Extensive experiments validate the effectiveness of our methods, demonstrating significant performance improvements in both segmentation and detection, especially in challenging unstructured scenarios.

## 1 Introduction

Autonomous driving technology has made significant strides in recent years, particularly in structured environments such as highways and urban roads. In these environments, the presence of clear lane markings, predictable traffic patterns, and standardized signage has allowed perception systems to achieve high levels of accuracy in tasks like lane detection, drivable area segmentation, and object detection. However, the transition to unstructured environments—characterized by irregular road layouts, diverse terrains, and a wide range of unpredictable traffic participants—presents a new set of challenges that current perception systems are not fully equipped to handle.

One of the critical challenges in advancing perception for unstructured environments is the availability of suitable datasets. While datasets like the Indian Driving Dataset (IDD) offer a rich array of images from unstructured scenarios, they often suffer from a lack of consistency in annotations across different tasks. Specifically, in the IDD dataset, the images annotated for semantic segmentation are not the same as those annotated for object detection. This non-overlapping nature of annotations reduces the efficiency of data utilization, as it limits the ability to jointly optimize multiple perception tasks using a single dataset. Additionally, there are several smaller datasets that are not as extensive as IDD, which collect data from unstructured areas such as rural and wilderness regions. While these datasets exhibit similar distributions, some contain segmentation annotations, while others include labels for depth estimation and object detection, illustrated in Figure 1, (a). The challenge lies in how to integrate these smaller datasets to enable mutual learning, allowing them to benefit from each other's labels and enhance their respective perceptual capabilities. Inspired by this challenge, this paper introduces the concept of Anti-Degradation Complementary Learning (ADC learning), a multi-stage semi-supervised multi-task training strategy that allows for the separate yet coordinated training of multi-tasks, thereby maximizing the use of available data.

Meanwhile, in unstructured environments, the complexity of the scene increases drastically. Roads may lack clear boundaries, obstacles can appear unexpectedly, and the types of objects encountered

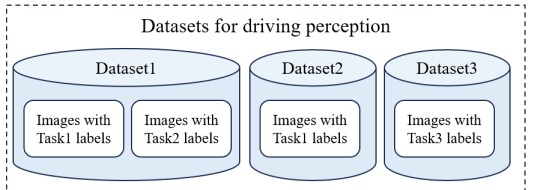 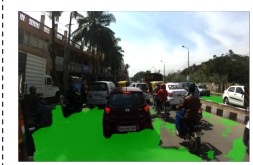 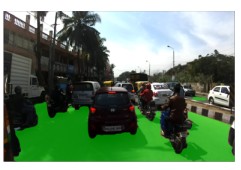

(a) The dataset challenge in unstructured scenarios.  (b) The fragmented phenomenon in complex scenarios.

Figure 1: The challenges in unstructured scenarios. (a) demonstrates the dataset challenge, and (b) shows the fragmented phenomenon, where the left side displays the predicted result and the right side displays the result processed by SAMEnhancer.

can vary widely. These factors complicate the perception tasks that are relatively straightforward in structured settings, necessitating more advanced and adaptable models. Moreover, the diversity and irregularity of unstructured environments mean that perception systems must be capable of handling a much broader range of scenarios, further increasing the difficulty of developing robust and reliable models. Segmentation based on densely annotated pixel points often lacks sufficient adaptability, resulting in fragmented representations of the road surface in complex environments, as shown in Figure 1, (b). To enhance the robustness and integrity of road recognition in unstructured scenes, this paper introduces the SAMEnhancer tool. SAM, derived from self-supervised training, possesses strong adaptability and generalization capabilities, along with a robust structural understanding of the data itself. SAMEnhancer utilizes a lightweight model, Mobile SAM, to integrate the strong semantic recognition of the original results with the robust structural properties of SAM, creating a plug-and-play lightweight tool that ensures structural integrity in recognition results within complex scenes.

Our contributions can be summarized as follows:

- We propose the Anti-Degradation Complementary learning (ADC learning) strategy that addresses the challenge of non-overlapping task annotations, enabling more efficient data utilization, especially in scenarios where the data is insufficient.
- We design a plug-and-play tool, SAMEnhancer, that leverages SAM to enhance the coherence and integrity of segmentation in complex scenes.
- We demonstrate the effectiveness of our approach through extensive experiments, showing significant improvements in drivable area segmentation and object detection in challenging unstructured scenarios.

## 2 RELATED WORK

### 2.1 MULTI-TASK LEARNING FOR PERCEPTION

Multi-task learning (MTL) has become a pivotal approach in computer vision, enabling the simultaneous learning of multiple tasks within a single framework. This strategy is particularly effective in perception systems, where tasks such as object detection, semantic segmentation, and depth estimation often share underlying features. Early work in MTL leveraged hard parameter sharing, where different tasks shared the same network backbone while maintaining task-specific output layers. Subsequent research explored soft parameter sharing, allowing each task to have its own set of parameters while still enabling information exchange across tasks, thereby mitigating task interference and enhancing learning flexibility (Lee & Liu, 2021).

Recent advancements have seen MTL applied to autonomous driving, where perception tasks must be performed in real-time and in diverse environments. LSNetDuan et al. (2021) consolidates object detection, instance segmentation, and pose estimation under the umbrella of location-sensitive visual recognition, employing a unified approach to tackle these tasks. Furthermore, Phillips et al. Phillips et al. (2021) explored deep multi-task learning for joint localization, perception, and prediction, highlighting the efficiency and accuracy gains achieved in complex real-world driving scenar-

ios. DLT-NetQian et al. (2019) adopts the encoder-decoder framework and actively constructs context tensors between subtask decoders, facilitating the sharing of specific information across tasks. MultiNetTeichmann et al. (2018) utilizes a shared encoder alongside three independent decoders to simultaneously address the three scene perception tasks: scene classification, object detection, and segmentation of the drivable area. In Wu et al. (2022), the YOLOP framework demonstrated the effectiveness of MTL in reducing computational costs and improving accuracy through an end-to-end network capable of simultaneously performing lane detection, object detection, and drivable area segmentation. This integration streamlined the processing pipeline and enhanced overall system performance. Their approach aimed to create more robust models by sharing knowledge between tasks such as object detection, semantic segmentation, and depth estimation. Lastly, the FULLER framework addressed limitations in existing multi-modal MTL methods in 3D autonomous driving scenarios, unifying multi-modality and multi-task learning through a multi-level approach, which is crucial for managing diverse data sources and tasks in autonomous driving Huang et al. (2023).

However, these approaches typically rely on well-aligned datasets with consistent annotations across all tasks, limiting their applicability in more complex scenarios like unstructured environments.

## 2.2 DATASETS FOR UNSTRUCTURED ENVIRONMENTS

To develop perception models suited for unstructured environments, it is essential to use datasets that accurately reflect the unique challenges of these settings. One such dataset is the Robot Unstructured Ground Driving (RUGD), which comprises over 7,000 RGB frames with pixel-wise annotations for semantic segmentation in off-road environments. Captured from a small unmanned mobile robot, RUGD presents challenging visual properties, including blurred frames and irregular class boundaries Wigness et al. (2019). Another noteworthy dataset is RELLIS-3D, collected on the Rellis Campus of Texas A&M University, which includes 13,556 LiDAR scans and 6,235 images across five traversal sequences on non-paved trails, documenting diverse environments like forests and pastures Duraisamy & Natarajan (2023). The ORFD dataset focuses on free space detection in various off-road scenes, such as forests and farmlands, under changing weather and lighting conditions Chang et al. (2007). The Freiburg Forest dataset, collected using a Viona autonomous mobile robot, features multi-spectral and multi-modal images at 20 Hz with a resolution of 1024x768 pixels, providing annotated pixel-wise segmentation masks for six classes Valada et al. (2017).

Among these datasets, the Indian Driving Dataset (IDD) Varma et al. (2019) stands out as one of the most comprehensive, capturing a wide array of scenarios, including rural roads, crowded city streets, and diverse weather conditions. This dataset provides annotations for semantic segmentation, object detection, and instance segmentation, making it a valuable resource for multi-task learning in unstructured environments. Data collection for IDD was conducted in India, where road conditions differ significantly from those in Europe and North America, featuring a diverse array of traffic participants, such as autorickshaws and animals. Compared to Cityscapes, IDD exhibits a notably different class distribution, with a higher proportion of motorcycles and two-wheeled vehicles. The diversity of background classes and ambient factors further contributes to the complexity of road scenes. While IDD offers 10,004 labeled images with finely delineated instance-level boundaries, its annotation types surpass those of Cityscapes in terms of object classes and appearance diversity.

Research by Baheti et al.Baheti et al. (2020) focuses on IDD, proposing modifications to the DeepLab V3+ Chen et al. (2018) framework by using lower atrous rates to improve dense traffic prediction, with a dilated Xception network serving as the backbone for feature extraction. These explorations lay the groundwork for achieving higher performance on the IDD dataset.

However, a notable limitation of the IDD dataset is the lack of consistency in annotations across tasks. Specifically, the segmentation and detection annotations are not provided for the same set of images, which complicates the training of joint models. This issue has prompted the need for novel training strategies, such as the alternating multi-task training proposed in this work, to fully exploit the potential of the IDD dataset.

## 2.3 SEGMENT ANYTHING MODEL (SAM) AND ITS APPLICATIONS

The Segment Anything Model (SAM) represents a significant leap forward in the field of image segmentation. Designed as a universal segmentation model, SAM can generate high-quality masks for

a wide variety of objects in diverse scenarios without the need for fine-tuning on specific datasets. It operates using a set of point prompts that guide the segmentation process, making it highly adaptable and effective in situations where traditional segmentation models may struggle. Recent advancements in SAM are highlighted in several studies, including the work by Kirillov et al. (2023), which emphasizes its capability to generate high-quality segmentation masks across various contexts Kirillov et al. (2023).

SAM has been successfully applied in various domains, from medical imaging to autonomous driving, where it has enhanced the precision of segmentation tasks. In the context of multi-task learning, SAM's ability to generate accurate masks from minimal input has the potential to significantly improve the performance of downstream tasks, such as object detection and segmentation. Zhang et al. (2023) provide a comprehensive survey on SAM, exploring its implications for vision and beyond, underscoring its versatility Zhang et al. (2023b). In this work, we integrate SAM into our multi-task perception framework, using it to generate masks that are fed into both segmentation and detection branches, thereby improving overall accuracy. Setu et al. (2024) further unveil SAM's functionality, applications, and practical implementation across multiple domains Setu et al. (2024).

However, SAM's high computational complexity leads to suboptimal performance in real-time applications. To address this issue, researchers have proposed a series of acceleration and optimization methods. First, some studies have introduced lightweight encoders and decoupled distillation techniques to significantly reduce the model's computational complexity and size, enabling fast and efficient image segmentation on mobile devices Zhang et al. (2023a). Additionally, other research has proposed new training strategies and hardware acceleration schemes, greatly enhancing segmentation speed to meet the demands of real-time applications Zhao et al. (2023). Furthermore, some studies have designed efficient visual transformer architectures that optimize model structure and parameter configurations, significantly increasing processing speed without compromising performance, making them suitable for resource-constrained environments Zhang et al. (2024).

These studies demonstrate that through optimization and acceleration techniques, SAM can achieve efficient operation in real-time applications while maintaining high accuracy. In summary, the powerful object generalization ability of SAM, combined with these optimization methods, provides robust support for feature representation and perception in complex scenes, thereby enhancing the overall effectiveness of perception models across diverse applications.

## 3 METHOD

This paper uses YOLOP as the foundational structure. Due to the diverse types of unstructured roads, which include dirt roads, concrete roads, and sandy roads, and the fact that lane markings may not always be present on the surface, so we remove the lane detection branch from YOLOP while retaining the branches for object detection and road segmentation. To enhance the utilization of unstructured road data, we propose a complementary learning approach that links two originally independent datasets for multi-task learning. To improve the accuracy and coherence of road segmentation, we introduce SAMEnhancer, which leverages the advantages of self-supervised learning to enhance segmentation results. The following sections will provide a detailed introduction.

### 3.1 ANTI-DEGRADATION COMPLEMENTARY LEARNING

In structured scenarios, a commonly employed strategy is multi-task learning techniques, as illustrated in Figure 2 (a). This approach aims to achieve the simultaneous execution of various tasks within a unified framework using different task data annotations based on the same dataset, thereby enhancing perceptual robustness and generalization. However, the data in unstructured scenarios is not as comprehensive as that in structured scenarios, and many datasets can only support single tasks. For example, R2D2 includes labels for object detection, ORFD contains labels for segmentation, and IDD features labels for image segmentation and localization, with no overlapping images corresponding to the labels. This leads to the following issues:

1. Low Data Utilization: A single image may contain information for all perceptual tasks, but insufficient annotations limit the extractable information from that image.

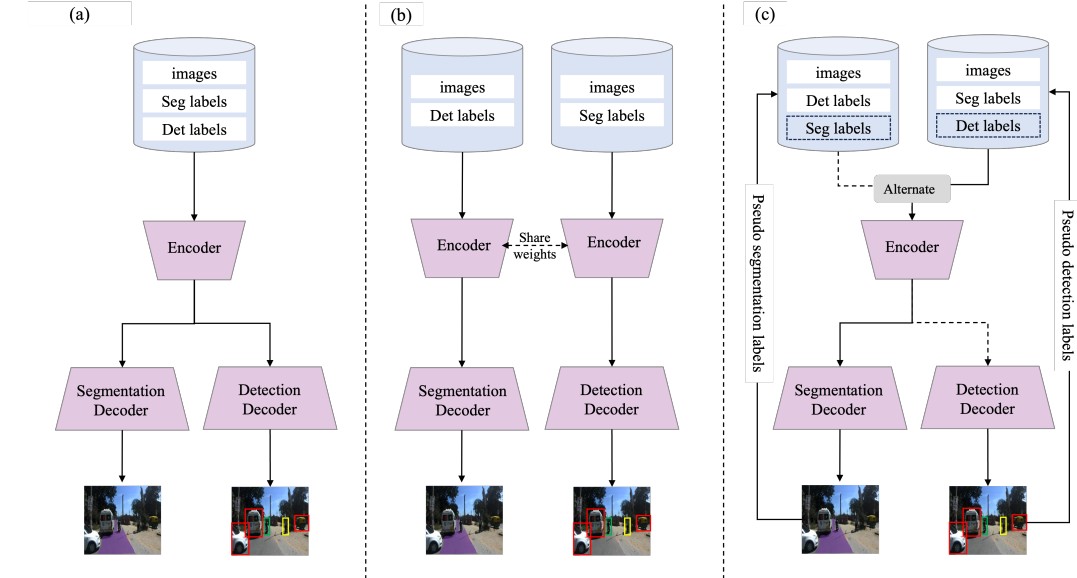

Figure 2: The contrast of different training strategies. (a) represents a multi-task training network with complete data labels, (b) illustrates collaborative training of two tasks through a shared encoder when data labels are incomplete, achieving the goal of multi-task learning, and (c) depicts the training strategy proposed in this study, where alternating training allows the two tasks to provide each other with pseudo-labels, realizing Anti-Degradation Complementary Learning.

2. Limited Data for Individual Tasks: The current scale of perception data in unstructured scenarios is relatively small, providing limited data for single tasks.

A straightforward strategy is to have the network alternately learn on two tasks, directly sharing weights in the feature extractor to enhance the network's cross-task capabilities, as depicted in Figure 2 (b). However, this strategy often encounters network degradation issues, meaning that after training on the next task, there is a significant drop in performance on the previous task.

To address the existing problems, this paper proposes a method called Anti-Degradation Complementary Learning (ADC learning) based on a multi-task learning framework, incorporating alternating learning strategies and semi-supervised learning, as shown in Figure 2 (c). ADC learning consists of two phases: Phase One involves simple alternating training as illustrated in Figure 2 (b), utilizing a shared encoder trained on their respective datasets. When training one task, the branch for the other task is frozen. This learning process runs for a limited number of epochs to acquire more generalized knowledge from the two tasks. Phase Two activates the pseudo-label training module, where in each training stage, one task is trained using ground truth labels while the other utilizes pseudo-labels. For each training stage involving pseudo-label training, the pseudo-labels are generated from the model trained on the ground truth in the previous phase, applied to the current task's dataset.

With the intervention of pseudo-label supervision, the network can continue learning the knowledge of the current task while retaining the knowledge gained in the previous phase, effectively preventing model degradation. This creates a semi-supervised multi-task complementary learning state where originally independent datasets can leverage each other's information to accomplish perceptual tasks that lack ground truth. This significantly enhances the utilization of data and labels while improving model generalization.

## 3.2 SAMEnhancer

The network training for the target road enables the model to have pixel-wise semantic discrimination capabilities, which, while sufficiently fine-grained, also presents an issue of inconsistency: pixels with the same semantics are often concentrated in certain regions, and pixel-level discrim-

**Algorithm 1:** Anti-Degradation Complementary Learning Process

```
   /* alternate training stage:                                      */
 1 begin
 2    mode = segmentation or detection;
 3    while epoch < mix_epoch do
 4       while i < alternate_frequency do
 5          freeze !mode decoder;
 6          input mode data;
 7          model.train();
 8          compute supervised loss;
 9       end while
10       switch mode;
11    end while
12 end
   /* Mix training stage:                                            */
13 begin
14    unfreeze all layers;
15    while epoch < end_epoch do
16       if !epoch % updata_frequency then
17          update pseudo labels
18       end if
19       input dataset with pseudo labels;
20       model.train();
21       compute supervised and semi-supervised loss;
22    end while
23 end
```

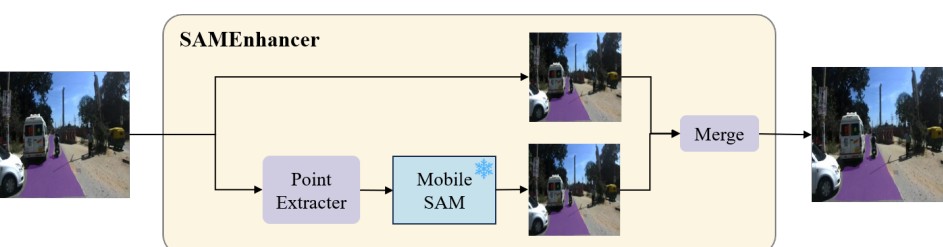

Figure 3: The process of the proposed SAMEnhancer algorithm.

ination weakens the structural judgment at the regional level, leading to isolated points and holes in the segmentation results. On the other hand, SAM, which learns in a self-supervised manner, lacks the guidance of dense labels and semantic information, but possesses strong generalization and adaptability, emphasizing the capture of structured information within the data. As a result, its segmentation results exhibit greater coherence and completeness. Based on this, this paper proposes SAMEnhancer, which optimizes segmentation results by combining the strong semantic capability of label-guided network training and the strong structural connectivity of the SAM model. The process is illustrated in Figure 3.

The SAMEnhancer takes the image $I$ and network segmentation result $\tilde{\mathbf{Y}}$ as input, $\tilde{\mathbf{Y}} \in \mathbb{R}^{C \times H \times W}$, $\tilde{\mathbf{Y}}_{c,h,w} \in [0, 1]$, which represents the confidence score. $C$ represents the number of categories; for road segmentation, it is set to 2. $H$ and $W$ represent the image height and width, respectively. After processing with SAMEnhancer, the tensor shape remains the same, and the values are binarized to 0 and 1. The SAMEnhancer operates through a three-stage process, which will be explained in the following sections.

### 3.2.1 POINT PROMPT EXTRACTION

The first step is to extract points, which serves to provide prompts for SAM in the subsequent steps. The prompts need to meet the following conditions: 1. Inside the target area, 2. Close to the center, and 3. With high confidence. Thus, we selected three points as the prompt point set: the centroid indicates the center of the polygon weighted by confidence; the center of the minimum enclosing circle indicates the morphological center of the polygon; to avoid the first two points falling outside the polygon, we add the point of highest confidence inside. The mathematical process is as follows:

Specifically, the output image $y^{pred}$ is first subjected to morphological transformations, performing opening followed by closing operations:

$$\text{morph}(\tilde{\mathbf{Y}}) = \text{close}(\text{open}(\tilde{\mathbf{Y}})) \tag{1}$$

which can remove isolated points in the prediction results and smooth the boundaries of regions, thereby avoiding noise from affecting keypoint extraction. The opening and closing operations are defined by equations 2 and 3, respectively:

$$\text{open}(\tilde{\mathbf{Y}}) = (\tilde{\mathbf{Y}} \ominus k^{open}) \oplus k^{open} \tag{2}$$

$$\text{close}(\tilde{\mathbf{Y}}) = (\tilde{\mathbf{Y}} \oplus k^{close}) \ominus k^{close} \tag{3}$$

where $\ominus$ denotes the erosion operation, $\oplus$ denotes the dilation operation, and $k$ represents the structuring element kernel. Then, find the polygon areas $A^P$ in the transformed prediction mask:

$$A^P = \text{findPoly}(\text{morph}(\tilde{\mathbf{Y}})) \implies \{A_1^P, A_2^P, \ldots, A_k^P\} \tag{4}$$

in which, $A^P$ represents $k$ binarized polygonal regions, and we use the prediction confidence $C$ of each pixel as the value of the polygon, resulting in:

$$V_i^P = C \times A_i^P \tag{5}$$

In each polygon $P_i$, three feature points are identified: the centroid $pt^{cent}$, the circumcircle center $pt^{exc}$, and the highest confidence internal point $pt^{hconf}$, forming the feature point set: $P_i = [pt_i^{cent}, pt_i^{exc}, pt_i^{hconf}]$. $pt_i^{cent}$ is defined as equation 6.

$$pt_i^{cent} = (x_i^{cent}, y_i^{cent}) = \left( \frac{\sum_{x=0}^{W-1} \sum_{y=0}^{H-1} x \cdot V_i^P(x,y)}{\sum_{x=0}^{W-1} \sum_{y=0}^{H-1} V_i^P(x,y)}, \frac{\sum_{x=0}^{W-1} \sum_{y=0}^{H-1} y \cdot V_i^P(x,y)}{\sum_{x=0}^{W-1} \sum_{y=0}^{H-1} V_i^P(x,y)} \right) \tag{6}$$

$pt^{exc}$ and $pt^{hconf}$ are the center of the minimum enclosing circle and the internal point with the highest confidence, as given in Equation 7 and Equation 8, respectively:

$$pt_i^{exc} = minEnclosingCircle(A_i^P) \tag{7}$$

$$pt_i^{hconf} = argmax(V_i^P) \tag{8}$$

Finally, remove the points that are not inside the polygon.

### 3.2.2 SEGMENTATION OPTIMIZATION VIA SAM

In the second stage, we utilize the extracted internal points as inputs to the mobile SAM. SAM is developed using self-supervised training, leveraging a large dataset of over 1 billion images. This extensive training allows SAM to achieve excellent generalization across diverse segmentation tasks. It can effectively utilize various prompts like points and masks, including points and bounding boxes, enabling users to interactively guide the segmentation process. However, SAM is relatively slow in terms of processing speed. Mobile SAM addresses this issue by enhancing speed while maintaining

the model's performance, making it more suitable for real-time applications. In this paper, we aim to achieve automatic extraction without human interaction by using the key points from the previous section as prompts input into Mobile SAM, thereby obtaining the generated masks $\hat{Y}$. The formula is shown in Equation 9.

$$\hat{\mathbf{Y}} = \text{mobile SAM}(I, P) \tag{9}$$

### 3.2.3 RESULT FUSION

The final stage of the SAMEnhancer process involves fusing the mobile SAM segmentation results with the original predictions from the network. This fusion is guided by confidence scores, in $\tilde{Y}$, the portions with confidence above the threshold (0.9 in this paper) are retained, while the other parts keep the results from $\tilde{Y}$, The fusion process can be mathematically represented as Equation 10.

$$Y^{\text{merge}} = \begin{cases} \tilde{Y} & \text{if } V^P > 0.9 \implies \text{High semantic confidence} \\ \hat{Y} & \text{if } V^P \le 0.9 \implies \text{Strong structural coherence} \end{cases} \tag{10}$$

By integrating the strengths of both the original and refined outputs, the SAMEnhancer generates a final segmentation map that maintains high accuracy and consistency, particularly in challenging scenarios with diverse road conditions.

In summary, the SAMEnhancer tool is designed to optimize segmentation outcomes by effectively combining the strengths of initial predictions with the results generated through mobile SAM. This approach not only enhances segmentation accuracy but also improves the overall performance of segmentation in unstructured autonomous driving applications.

## 4 EXPERIMENTS AND RESULTS

### 4.1 EXPERIMENT SETTING

A series of experiments were conducted in this study to validate the effectiveness of ADC learning and SAMEnhancer. The datasets used were IDD and Bdd100K. For the object detection task, the objects were reclassified into five categories: motor vehicles, non-motor vehicles, people, animals, and traffic signs. For the drivable area, no distinction was made between road and drivable fall-back. The experiments were based on the PyTorch framework, utilizing torch 2.0.1+cu117 and a GeForce 4090 GPU. The experimental setup was based on the YOLOP network architecture, with the lane detection branch removed. The following sections will provide a detailed description of the validation experiments for ADC learning and SAMEnhancer.

### 4.2 RESULTS

Figure 4 presents the visualization results of the proposed method based on YOLOP. The first row shows the results of training the IDD dataset using only alternate training, the second row presents the outcomes of training with ADC learning, and the third row presents the results after processing with SAMEnhancer. It can be observed that the ADC learning strategy reduces the occurrence of perception errors, while SAMEnhancer extracts a more coherent and complete drivable surface on this basis.

### 4.3 ADC LEARNING

This study first validated that the performance of the model declines when using non-overlapping annotated data solely with rotation training, as shown in Table 1. The Bdd100K dataset was utilized, which is a city road dataset suitable for multi-task learning. The complete dataset was used for experiments and then divided into two parts: one with segmentation labels and the other with detection labels, simulating the scenario of non-overlapping annotations. Subsequently, ADC learning was applied to this data labeling, resulting in the outcomes shown in Table 1. Recall and mAP are evaluation metrics for object detection, while mIoU is an evaluation metric for road segmentation.

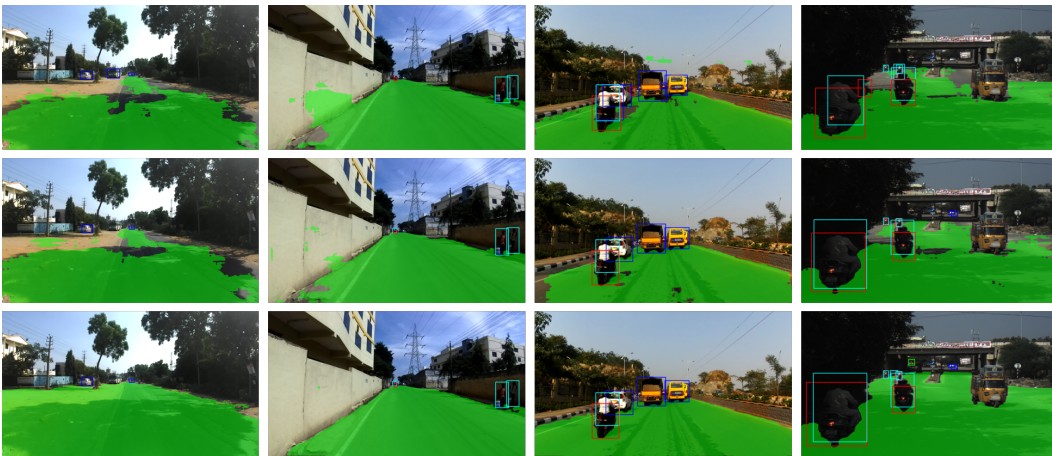

Figure 4: The visualization experiment results. The top row shows the results of YoloP on IDD dataset without ADC learning, the middle row presents the result with ADC learning, and the bottom row shows the results with ADC learning and SAMEnhancer.

It can be observed that the model's performance decreased under non-overlapping annotations, but improved after employing the proposed ADC learning. A similar phenomenon was also observed in the IDD dataset with the same experiment.

Secondly, this study validated the effectiveness of ADC learning under different network architectures. The encoder of YOLOP was replaced with ConvNeXt, EfficientNet, and DenseNet, respectively. The experimental results are shown in Table 2, demonstrating that ADC learning enhances model performance across various network structures.

Table 1: Ablation study on ADC learning with Bdd100K and IDD dataset. Recall and mAP50 are detection metrics, and mIOU is the segmentation metric.

|  | Bdd100KYu et al. (2020) | | | IDDVarma et al. (2019) | | |
|---|---|---|---|---|---|---|
|  | Recall | mAP50 | mIOU | Recall | mAP50 | mIOU |
| All label | 89.2 | 76.5 | 91.5 | - | - | - |
| Non-overlap label | 86.4 | 73.3 | 88.8 | 52.8 | 25.4 | 91.3 |
| ADC learning | 87.1 | 74 | 88.9 | 55.2 | 29.3 | 93.8 |

Table 2: Ablation study on ADC learning with different encoder architectures.

|  | witout ADC learning | | | with ADC learning | | |
|---|---|---|---|---|---|---|
| Encoder | Recall | mAP50 | mIOU | Recall | mAP50 | mIOU |
| ConvNeXtLiu et al. (2022) | 52 | 24.8 | 88.9 | 53.8 | 28.8 | 91.6 |
| EfficientNetKoonce & Koonce (2021) | 51.9 | 25.4 | 90.4 | 53.7 | 29.1 | 92.3 |
| DenseNetHuang et al. (2018) | 47.6 | 22 | 86.2 | 50.3 | 26.2 | 90.5 |
| CSPDarknet(YoloP)Wu et al. (2022) | 52.8 | 25.4 | 91.3 | 55.2 | 29.3 | 93.8 |

## 4.4 SAMENHANCER

This subsection tested the effects of using SAMEnhancer versus not using it. Table 3 presents the experimental results. In the table, ACC refers to accuracy, IoU indicates the performance of individual categories within the drivable area, and mIoU represents the average value between the drivable area and the background. The first row, "Network prediction," lists the metrics for the network's prediction results, the second row, "Mobile SAM prediction," shows the metric calculations for the

output values from the SAMEnhancer sampled points and mobile SAM predictions, and the third row, "Merge result" represents the final output after combining both results.

Table 3: The ablation study on SAMEnhancer. The top row shows the network output results, the middle row demonstrates the mobile SAM output results, and the bottom row depicts the final merged results.

|  | ACC | IOU | mIOU |
|---|---|---|---|
| Network Prediction | 96.9 | 89.5 | 93.8 |
| Mobile SAM Prediction | 96 | 88 | 91.2 |
| Merge result | 97.3 | 91.6 | 94.5 |

## 5 CONCLUSION

This study investigates multi-task learning in unstructured scenes and proposes Anti-Degradation Complementary learning (ADC learning) and SAMEnhancer. ADC learning addresses the scattered nature of unstructured scene datasets by designing a two-phase training method with joint non-overlap annotations, combined with semi-supervised learning. SAMEnhancer enhances the recognition performance of roads in unstructured scenes by leveraging the strong structural understanding of SAM along with the semantic understanding capability of the network itself, thereby addressing the fragmentation issue of roads in unstructured scenes. Qualitative and quantitative experiments demonstrate the effectiveness of the proposed methods.

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
