# OpenReview forum: "Multi-Task Perception in Unstructured Environments: Anti-Degradation Complementary Learning and SAMEnhancer"
_ICLR.cc/2025/Conference — ICLR 2025 Conference Withdrawn Submission_

### Official Review · Reviewer_pPqK · 2024-10-19

**Soundness:** 2
**Presentation:** 2
**Contribution:** 2
**Rating:** 3
**Confidence:** 5

**Summary:**

This paper presents an anti-degradation complementary learning approach for multi-task learning in unstructured scene understanding. A SAMEnhancer is introduced to improve segmentation quality. Experiments are conducted on IDD and BDD datasets, which verify the effectiveness of the proposed solution.

**Strengths:**

1. The proposed method addresses the challenging multi-task perception in unstructured environments.

2. Experiments on BDD and IDD datasets verify the effectiveness of the proposed approach.

**Weaknesses:**

1. Please consider comparing your proposed solution against more state-of-the-art multi-task perception methods such as YOLOP, YOLOPv2, HybridNets, etc. For example, their accuracy and efficiency results on BDD and IDD datasets could be added in the comparison tables to illustrate the superiority of your solution.

2. Please consider comparing your proposed SAMEnhancer and some existing SAM adapter or enhancement methods to verify the superiority of your proposal. For example, the Sam-adapter, Medical sam adapter, Classwise-sam-adapter, and PA-SAM could be compared.

3. The computational complexity and efficiency results like FLOPs/MACs, the number of parameters, memory requirements, and the running time of your model should be discussed, which are relevant for autonomous driving applications, especially in unstructured environments.

4. As shown in Fig. 4, some of the predictions at the road boundaries and shallows in the visualized examples of unstructured road environments are still not so precise, which could be discussed and analyzed in detail. How can the segmentation in such scenarios be further enhanced?

**Questions:**

1. Would you consider discussing some limitations of the presented work and pointing out some future research directions?

2. Would you consider providing more detailed ablations or parameter studies to help understand the gain of each component in your presented framework? For example, the threshold for fusion and the amount of data used for the semi-supervised training can be analyzed in the experiments.

---

### Official Review · Reviewer_Skkv · 2024-11-01

**Soundness:** 2
**Presentation:** 2
**Contribution:** 1
**Rating:** 3
**Confidence:** 5

**Summary:**

The manuscript presents two practical techniques that can be used to improve models for scene understanding. The first technique (ADC) improves the training of a multi-modal model when different folds are annotated for different modalities. The authors propose to train with mixed batches in which the labeled modalities are trained with full supervision while the unlabeled modalities are trained on pseudo labels. Experimental section shows improved performance of model for simultaneous semantic segmentation and object detection. The second technique leverages an external segmentation model to improve the quality of the recovered regions. The proposed approach consists of three stages as follows: i) representing polygonal regions with a set of points, ii) segment the selected points with Mobile SAM and iii) merge the original segmentation with SAM output according to prediction accuracy.

**Strengths:**

- the manuscript addresses an important application field of computer vision

- the proposed techniques improve the generalization performance

**Weaknesses:**

- in both cases, we observe improvement due to training on more data (unlabeled fold or SA-1B), which does not come up as a surprise

- many semi supervised approaches have been proposed in earlier work so the authors can not claim novelty

**Questions:**

Please note that SAM has not been trained with self supervision. Instead, it has been trained on a fully labeled dataset SA-1B. The manuscript could be improved by providing a better description of the SAM's training process and by discussing how the SA-1B dataset complements the proposed learning setup.

Postprocessing with mobile SAM is related to postprocessing with CRF as proposed in the first DeepLab paper (PAMI 2018). The manuscript could be improved by providing a more detailed comparison between the SAMEnhancer and the CRF postprocessing. Specifically, it would make sense to compare Mobile SAM with CRF in terms of generalization performance and computational efficiency.

---

### Official Review · Reviewer_GkDA · 2024-11-02

**Soundness:** 1
**Presentation:** 1
**Contribution:** 1
**Rating:** 1
**Confidence:** 5

**Summary:**

The paper proposes an approach to jointly learn NN representations for different tasks such as detection and segmentation using different non-overlapping datasets. The paper proposes to alternatingly optimize the task heads, along with pseudo labeling for the complementary task corresponding to each dataset for which annotations are not available. In addition, there is a SAM Enhancer module to use knowledge from pretrained SAM model.

In terms of experiments, the paper presents results of their method on the IDD_BDD combo, along with ablations on their method.

**Strengths:**

1. The paper is of practical significance
2. The method presented in the first part in 3.1 is simple, logical and makes sense.
3. The paper, until section 3, is well written.

**Weaknesses:**

1. Method:

a. The paper alternatingly optimizes for the different tasks. How is that different from joint training? The encoder is still being jointly trained, the task heads are frozen, which makes sense
b. Pseudo labeling can lead to a lot of erroneous predictions, there needs to be confidence filtering
c. I am not sure I understand the strong structural confidence claim in section 3.2.3
d. The method for point prompt extraction seems quite intricate, but I don't understand the need for that complexity. SAM provides far higher quality pseudo labels than the pseudo labeling in the preceding section where there is no filtering.

2. Experiments

This is a key flaw with this paper, the experiments are extremely incomplete. There are absolutely no comparisons with related work, and multi-task learning, representation learning, etc are well studied problems.

3. References

I see that a lot of relevant references on IDD, multi-task learning, representation learning are missing.

Overall, the method seems ad-hoc, the experiments are incomplete and the writing is not well done (esp after section 4 and references). It appears to be rushed and incomplete submission.

**Questions:**

I combined the weaknesses with the questions.

I realize it is too late to make any changes about the method.

Experiments - more datasets and tasks are needed, also, at this point I see only one proof of concept experiment on IDD, which is not sufficient for the exhaustive evaluation of the method for a complete paper.

There are absolutely no comparisons!!

Again, i feel it may not be possible to conduct so many experiments during the rebuttal period of 1 week...

Please add all relevant references - it is extremely incomplete.

The paper appears to be a rushed and an incomplete submission.

---

### Official Review · Reviewer_FchQ · 2024-11-04

**Soundness:** 1
**Presentation:** 2
**Contribution:** 1
**Rating:** 1
**Confidence:** 5

**Summary:**

This paper identifies the problem of multi-task perception in unstructured environments, noting that available datasets in this setting often have labels for differing tasks (e.g. detection, segmentation) provided on completely non-intersecting data samples. As a result, training perception models in conjunction for differing tasks is challenging. To address this challenge, the paper proposes two novel methods: Anti-Degradation Complementary (ADC) Learning and SAMEnhancer. ADC learning functions through alternating training on each perception task between using pseudo-labeled and human-labeled data per epoch. SAMEnhancer uses a lightweight version of SAM (Mobile SAM), which prompted with selected points, generates another segmentation mask which is combined with the output from the trained segmentation model to enhance results on low-confidence pixels.

**Strengths:**

1. The method is simple and easy to implement.
2. The setting is a novel one I have not seen previously considered.

**Weaknesses:**

1. The novelty of the approach is minimal. The key contributions of the paper are mostly direct combinations of already-existing methods (e.g. self-training with pseudo-labels and SAM) with no additional analysis. Furthermore I suspect similar approached which use some sort of SAM ensemble likely already exist.
2. The experimental results are completely inadequate. No ablation studies are performed on the key components of the approach, and no comparison with prior art (or other compelling baselines) is included.
3. Improvements of the proposed methods over simply using non-overlapping training data are marginal at best.
4. The methods section is poorly structured and the method is not well explained. For example, the motivation for the idea behind the SAMEnhance approach is not clear to me.
5. The prior work section is inadequate, and should include literature on semi-supervised learning approaches and other methods incorporating SAM into domain-specific perception pipelines.

**Questions:**

1. I do not understand the point of prompting the Mobile SAM model with specific points. What is the purpose of doing this? Is it supported with experiments?
2. Why are there no experiments combining both ADC Learning and SAMEnhancer? The two methods are compatible with each other, so I'm confused why they are treated as completely different parts of the story.
3. How does ADC Learning compare to simply performing semi-supervised learning without the alternating pseudo-label training? This seems like an obvious baseline for comparison.

---

### Note · Authors · 2024-11-25

I have read and agree with the venue's withdrawal policy on behalf of myself and my co-authors.